

# Integrating multimodal clinical data to predict intravenous (IV) fluid utilization: a comparative analysis of natural language processing techniques

Hairong Wang[1,2], Haipeng Ling[3] and Xingyu Zhang[4]

[1] College of Computing, Georgia Institute of Technology, Atlanta, GA, United States
[2] Department of Civil and Environmental Engineering, Carnegie Mellon University, Pittsburgh, PA, United States
[3] Business Analytics Program, Carnegie Mellon University, Pittsburgh, PA, United States
[4] Department of Communication Science and Disorders, School of Health and Rehabilitation Sciences, University of Pittsburgh, Pittsburgh, PA, United States

## ABSTRACT

**Background:** Accurate prediction of intravenous (IV) fluid utilization in emergency departments (ED) is essential for optimal clinical decision-making, resource allocation, and patient management. However, existing predictive models typically rely solely on structured clinical variables, overlooking the rich, contextual insights available in unstructured patient narratives. This study aimed to develop and evaluate predictive models that integrate structured clinical data with natural language processing (NLP)-derived embeddings from unstructured patient narratives to enhance prediction accuracy for IV fluid administration in ED settings.

**Methods:** We analyzed a large dataset from the National Hospital Ambulatory Medical Care Survey—Emergency Department (NHAMCS-ED, $n$ = 13,115), comprising both structured patient demographics and clinical variables, alongside unstructured chief complaints. Patient narratives were processed using three distinct Natural Language Processing (NLP) techniques: CountVectorizer, Word2Vec embeddings, and pre-trained GPT-2 embeddings. An early fusion strategy was employed, concatenating structured and unstructured features *before* feeding them into Logistic Regression (LR) and Gradient Boosting Classifier (GBC) models. Model performance was evaluated using 5-fold cross-validation, with the area under the receiver operating characteristic curve (AUC) as the primary metric.

**Results:** The integrated models, which combined structured clinical variables with NLP features from patient narratives, consistently demonstrated superior predictive performance over models using either data type alone. We compared several NLP techniques, including CountVectorizer, Word2Vec, and pre-trained Generative Pre-trained Transformer 2 (GPT-2) embeddings. Unexpectedly, the integrated models using traditional NLP features, particularly CountVectorizer, achieved the best performance, reaching an AUC of 0.786 with the Gradient Boosting Classifier (GBC) and 0.781 with logistic regression. These results outperformed the integrated model using GPT-2 embeddings (GBC AUC = 0.772). This suggests that for the short, keyword-driven patient complaints found in the emergency department, frequency-based methods may be more effective at extracting predictive signals.

**Conclusions:** Our findings confirm that integrating patient narratives with structured clinical data significantly enhances predictive accuracy for IV fluid

Corresponding authors
Hairong Wang,
hairongw@andrew.cmu.edu
Xingyu Zhang, xiz261@pitt.edu

utilization. This multimodal approach, particularly when leveraging traditional NLP models like CountVectorizer, provides a robust framework for improving clinical decision-making, resource allocation, and patient care in the ED. The study highlights the critical importance of aligning NLP model choice with the specific characteristics of clinical text data.

## INTRODUCTION

Intravenous (IV) fluid therapy (*Hoste et al., 2014*; *Myburgh & Mythen, 2013*) is commonly administered in emergency departments (EDs), playing a critical role in managing conditions ranging from mild dehydration to severe, life-threatening illnesses requiring urgent fluid resuscitation. Despite its routine use, variability and uncertainty persist regarding IV fluid administration decisions, potentially leading to inconsistent patient care, unnecessary resource utilization, and adverse patient outcomes (*Pines, Griffey & Cone, 2015*). Optimizing the decision-making process for IV fluid use can substantially enhance clinical effectiveness, patient safety, and operational efficiency (*Kawamoto et al., 2005*).

Advancements in medical informatics, particularly the widespread implementation of electronic health records (EHRs), have facilitated access to extensive datasets comprising both structured clinical variables and unstructured patient narratives (*Hripcsak & Albers, 2013*; *Jensen, Jensen & Brunak, 2012*). Structured data typically include quantifiable metrics such as demographic information, vital signs, medical history, and clinical severity indicators. While structured data offer easily interpretable and standardized measures, they often fail to capture the complete clinical context required for precise patient management decisions. In contrast, unstructured data, such as narrative text documented by clinicians during patient visits, provide richer context and nuanced details that are often absent from structured data (*Meystre et al., 2017*; *Xiao, Choi & Sun, 2018*).

Recent developments in natural language processing (NLP), especially transformer-based models such as Generative Pre-trained Transformer 2 (GPT-2), have revolutionized the extraction and interpretation of critical medical insights from unstructured textual data (*Kreimeyer et al., 2017*; *Lee et al., 2020*; *Radford et al., 2019*). These advanced NLP methods enable sophisticated understanding and quantification of patient narratives, which can significantly augment clinical predictions derived from structured data alone (*Esteva et al., 2019*; *Wu et al., 2020*). Indeed, prior research has consistently demonstrated the value of combining these data modalities for a range of clinical predictions, from inpatient mortality and 30-day readmissions (*Rajkomar et al., 2018*) to predicting patient disposition from the ED (*Hong, Haimovich & Taylor, 2018*). These studies affirm that unstructured narratives contain diagnostic and prognostic

information often absent from structured fields alone. However, while the general benefit of data fusion is recognized, its application to resource-specific predictions like IV fluid utilization remains less explored. Furthermore, which NLP techniques are most effective for extracting signals from the short, keyword-driven texts common in the ED is an area requiring further investigation.

This study aimed to address this gap by developing and evaluating predictive models of IV fluid utilization in emergency departments using structured clinical data and NLP-derived insights from unstructured patient narratives. The primary objective was to assess whether an integrated predictive modeling approach, combining structured and unstructured data, would yield superior performance compared to models relying exclusively on either data type. Such models could inform clinical decisions, enhance patient care management, and optimize resource allocation in emergency medical practice.

## METHOD

The overall pipeline of our study is illustrated in Fig. 1. The process begins with data acquisition from the NHAMCS-ED dataset, followed by cleaning and a train/test split. The data is then separated into two streams: a structured data stream and an unstructured text stream. For the text data, we extracted features using three distinct NLP techniques: CountVectorizer, Word2Vec, and pre-trained GPT-2 embeddings. Finally, both single-modality (structured-only, unstructured-only) and integrated (combined *via* early fusion) datasets were used to train and evaluate Logistic Regression (LR) and Gradient Boosting Classifier (GBC) models for predicting IV fluid administration.

### Computing infrastructure

This study was conducted using a standard Google Colab environment. The experiments were performed on a typical Ubuntu-based system with available CPU and GPU resources provided by Colab, without the use of any specialized hardware. This setup was adequate for executing data preprocessing and model training tasks efficiently.

### Reproducibility and code repository

All experiments and analyses presented in this study are fully reproducible using the provided code and data. The complete implementation—including data preprocessing, model training, evaluation, and result visualization—is documented in a series of Jupyter notebooks designed to run on Google Colab. Detailed instructions, including environment setup and dependency installation, are provided in the repository's README file. The code leverages Python libraries such as scikit-learn, pandas, numpy, and transformers, ensuring that researchers can replicate the experiments using publicly available tools and datasets. The repository, along with the necessary datasets and instructions, is available at https://doi.org/10.5281/zenodo.17116941.

### Data preprocessing

This study conducted a retrospective analysis using data from the 2021 National Hospital Ambulatory Medical Care Survey–Emergency Department (NHAMCS-ED)

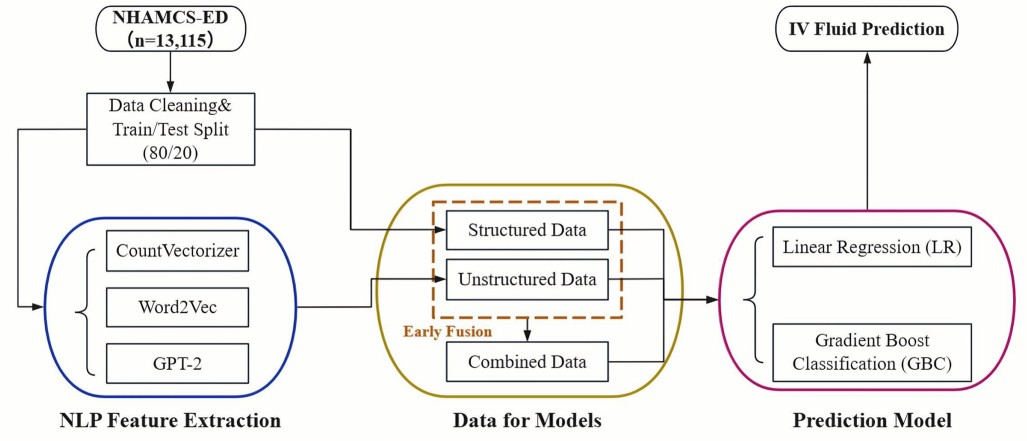

**Figure 1 Overall pipeline for predicting IV fluid administration in the ED.**

(_Cairns & Kang, 2022_), a nationally representative database of ED visits across the United States. The analytic sample included 13,115 adult patients aged 18 years or older after excluding pediatric cases. The dataset contained structured data—including demographic details, clinical characteristics, and medical histories—as well as unstructured data comprising free-text patient narratives documenting chief complaints and injury reasons.

Structured data variables consisted of demographics (age, sex, race/ethnicity), visit information (arrival time, mode of transportation, visit type), clinical measurements (temperature, heart rate, respiratory rate, pulse oximetry, pain levels), and the Emergency Severity Index (ESI) score. Medical history included chronic conditions such as Alzheimer's disease, diabetes, chronic obstructive pulmonary disease, and coronary artery disease. Additional structured factors were residence type, insurance type, presence of injury, trauma, poisoning, or adverse medical effects. Missing data was addressed using median imputation. Unstructured data encompassed free-text patient-reported chief complaints and descriptions of injuries, providing additional clinical insights beyond structured metrics. NLP techniques were utilized to transform narrative text into meaningful predictive variables.

The study population was characterized using descriptive statistics stratified by IV fluid administration status. Frequencies and percentages were reported for categorical variables, and comparisons were conducted using chi-square tests. Multivariable logistic regression was performed to identify independent predictors of IV fluid utilization, employing a stepwise variable selection process to enhance model interpretability and reduce multicollinearity. Statistical significance was defined at a $p$-value of <0.05, with all hypothesis tests conducted as two-sided analyses.

### Natural language processing feature extraction

**CountVectorizer:** We implemented a bag-of-words (BoW) model using scikit-learn's CountVectorizer. The raw text of patient complaints was first preprocessed by converting to lowercase and removing special characters. We configured the vectorizer to build a

vocabulary of the top 500 features based on term frequency across the *corpus*, considering both unigrams and bigrams (ngram_range = (1, 2)). To ensure robustness, terms that appeared in fewer than two documents (min_df = 2) or in more than 95% of documents (max_df = 0.95) were excluded. English stop words were also removed.

**Word2Vec:** To capture semantic relationships between words, we trained a custom Word2Vec model using the gensim library. The narrative texts were tokenized and fed into a Continuous Bag-of-Words (CBOW) model (sg = 0). The model was configured with a vector size of 100, a window size of 5, and a minimum word count of 2. The model was trained for 10 epochs. For each patient narrative, a single feature vector was generated by averaging the vectors of all words present in that text and in the model's vocabulary.

**GPT-2 embeddings:** For a transformer-based representation, we utilized a pre-trained GPT-2 model from the transformers library. Each patient narrative was tokenized and processed in batches to generate embeddings. We extracted the hidden state of the first token (often representing the (CLS) token's aggregated sentence meaning) from the last layer of the model as the fixed-size embedding for each text.

## Predictive modeling and integration strategy

Predictive models were developed using structured and unstructured data separately and subsequently integrated through an early fusion approach. Structured data models included LR (*Stoltzfus, 2011*) and GBC (*Chen & Guestrin, 2016*; *Natekin & Knoll, 2013*), with hyperparameter tuning to optimize performance parameters such as tree depth, learning rate, and number of estimators. For unstructured data, narrative entries from patient records were processed using the Generative Pre-trained Transformer 2 (GPT-2) model, generating numerical embeddings that were inputs for logistic regression and GBC models. For the integrated models, we employed an early fusion approach where the normalized structured features were horizontally concatenated with the NLP-derived text features (from CountVectorizer, Word2Vec, or GPT-2, respectively) to create a single, unified feature matrix for each NLP method. These combined matrices were then used as input for the LR and GBC models. Model performance was assessed through 5-fold cross-validation to ensure robustness and reduce potential overfitting.

## Evaluation method

Evaluation metrics included accuracy, sensitivity, specificity, and the area under the receiver operating characteristic (ROC) curve (AUC). The optimal decision thresholds were identified based on ROC analysis, balancing clinical sensitivity and specificity (*Steyerberg et al., 2010*).

## Study design and ethical considerations

This research utilized publicly available, anonymized data from NHAMCS-ED, managed by the U.S. Centers for Disease Control and Prevention. As this secondary analysis involved de-identified data, the study received exempt approval from the Institutional

Review Board (IRB) at the University of Pittsburgh (protocol STUDY24120115), adhering to ethical guidelines for secondary data use.

## RESULTS

A total of 13,115 adult patients were included in this analysis, of which 4,193 (32.0%) received intravenous (IV) fluids during their ED visits (Table 1). Patients who received IV fluids were more frequently female (56.1% *vs.* 53.0%, $p = 0.0008$), older (28.8% aged ≥65 years *vs.* 19.6%, $p < 0.0001$), and predominantly White (65.1% *vs.* 56.1%, $p < 0.0001$). They were also more likely to reside in nursing homes (3.4% *vs.* 1.5%, $p < 0.0001$) and had higher proportions of Medicare insurance coverage (33.9% *vs.* 23.7%, $p < 0.0001$). Clinically, IV fluid recipients showed greater severity indicators, such as higher rates of ambulance arrival (29.2% *vs.* 17.3%, $p < 0.0001$), severe or unbearable pain levels (31.1% *vs.* 28.1%, $p = 0.0187$), elevated body temperature above 38°C (3.6% *vs.* 1.1%, $p < 0.0001$), increased heart rates greater than 90 bpm (44.7% *vs.* 36.0%, $p < 0.0001$), lower oxygen saturation levels (15.2% *vs.* 8.2% with pulse oximetry ≤94%, $p < 0.0001$), and elevated respiratory rates greater than 20 breaths per minute (14.1% *vs.* 6.6%, $p < 0.0001$). Additionally, IV fluid recipients more often presented with conditions related to overdose, poisoning, adverse medical effects, or trauma (21.6% *vs.* 36.2%, $p < 0.0001$) and had higher Emergency Severity Index scores categorized as immediate or emergency (28.4% *vs.* 14.3%, $p < 0.0001$). Patients receiving IV fluids also had notably higher prevalence of chronic conditions, including coronary artery disease (12.4% *vs.* 5.7%, $p < 0.0001$), diabetes mellitus type II (12.6% *vs.* 7.9%, $p < 0.0001$), hypertension (41.8% *vs.* 28.5%, $p < 0.0001$), chronic kidney disease (7.7% *vs.* 3.5%, $p < 0.0001$), and obesity (10.3% *vs.* 7.5%, $p < 0.0001$).

Figure 2 presents a heatmap visualizing the percentage point differences in demographic and clinical characteristics between patients who received IV fluids and those who did not. This visualization highlights the most significant differentiating factors. Notably, patients with an 'Urgent' or 'Semi-urgent' Emergency Severity Index score, those with a history of hypertension or diabetes, and those arriving by ambulance showed a markedly higher prevalence in the IV fluid group. Conversely, patients presenting with trauma or injury were substantially less likely to receive IV fluids, likely because their treatments were focused on other interventions. This graphic representation complements the detailed statistics provided in Table 1.

Figure 3 presents a forest plot illustrating odds ratios (ORs) and corresponding 95% confidence intervals (CIs) for significant predictors of IV fluid utilization derived from multivariable logistic regression analysis. The plot visually depicts the relative influence of various patient demographics, clinical characteristics, and medical histories on the likelihood of receiving IV fluids in emergency department visits. Key findings include notably higher odds for IV fluid administration among patients arriving by ambulance (OR = 1.99, 95% CI [1.82–2.17]), those aged ≥65 years (OR = 2.05, 95% CI [1.86–2.25]), patients with elevated temperatures (>38 °C; OR = 3.27, 95% CI [2.52–4.25]), and elevated respiratory rates (>20 breaths per minute; OR = 2.41, 95% CI [2.13–2.73]). Additionally, patients categorized with higher Emergency Severity Index (immediate or emergency) and those experiencing injury or adverse medical effects also exhibited significantly increased

**Table 1 Demographic and clinical characteristics of emergency department patients categorized by IV Fluid utilization.**

| | IV fluid | | |
| --- | --- | --- | --- |
| | **No** | **Yes** | ***p* value** |
| | 8,922 (68.0%) | 4,193 (32.0%) | |
| **Gender** | | | 0.0008 |
| Female | 4,728 (53%) | 2,354 (56.1%) | |
| Male | 4,194 (47%) | 1,839 (43.9%) | |
| **Age, y** | | | <0.0001 |
| 18–39 | 3,864 (43.3%) | 1,295 (30.9%) | |
| 40–65 | 3,312 (37.1%) | 1,692 (40.4%) | |
| >=65 | 1,746 (19.6%) | 1,206 (28.8%) | |
| **Race/ethnicity** | | | <0.0001 |
| White | 5,008 (56.1%) | 2,729 (65.1%) | |
| Black | 2,140 (24%) | 883 (21.1%) | |
| Hispanic | 1,393 (15.6%) | 434 (10.4%) | |
| Other | 381 (4.3%) | 147 (3.5%) | |
| **Residence type** | | | <0.0001 |
| Private residence | 8,171 (94.2%) | 3,881 (94.6%) | |
| Nursing home | 128 (1.5%) | 138 (3.4%) | |
| Homeless | 264 (3%) | 50 (1.2%) | |
| Other | 111 (1.3%) | 33 (0.8%) | |
| **Insurance type** | | | <0.0001 |
| Private insurance | 2,264 (28.1%) | 1,218 (31.9%) | |
| Medicare | 1,905 (23.7%) | 1,296 (33.9%) | |
| Medicaid or CHIP | 2,854 (35.4%) | 945 (24.7%) | |
| Uninsured | 686 (8.5%) | 255 (6.7%) | |
| Other | 345 (4.3%) | 105 (2.7%) | |
| **Day of week** | | | 0.0814 |
| Weekdays | 6,626 (76.2%) | 3,184 (77.6%) | |
| Weekend | 2,071 (23.8%) | 919 (22.4%) | |
| **Arrival time** | | | 0.7501 |
| Morning | 2,480 (28.6%) | 1,207 (29.3%) | |
| Afternoon | 2,718 (31.3%) | 1,262 (30.6%) | |
| Evening | 1,372 (15.8%) | 640 (15.5%) | |
| Night | 2,104 (24.3%) | 1,014 (24.6%) | |
| **Arrive by ambulance** | | | <0.0001 |
| Yes | 1,496 (17.3%) | 1,198 (29.2%) | |
| No | 7,169 (82.7%) | 2,898 (70.8%) | |
| **Follow-up visit** | | | <0.0001 |
| No | 7,373 (91.7%) | 3,682 (93.8%) | |
| Yes | 667 (8.3%) | 245 (6.2%) | |
| **Seen within last 72 h** | | | 0.7539 |
| Yes | 338 (4.3%) | 164 (4.1%) | |

| | IV fluid | | |
| --- | --- | --- | --- |
| | **No** | **Yes** | ***p* value** |
| No | 7,553 (95.7%) | 3,796 (95.9%) | |
| **Pain level** | | | 0.0187 |
| No or mild pain | 2,111 (39%) | 1,013 (37.7%) | |
| Moderate or severe pain | 1,775 (32.8%) | 836 (31.1%) | |
| Very severe and unbearable pain | 1,521 (28.1%) | 836 (31.1%) | |
| **Temperature** | | | <0.0001 |
| 36°C – 38°C | 7,950 (95.6%) | 3,680 (92.8%) | |
| <= 36°C | 273 (3.3%) | 142 (3.6%) | |
| > 38°C | 95 (1.1%) | 142 (3.6%) | |
| **Heart rate, times/min** | | | <0.0001 |
| <= 60 | 381 (4.6%) | 181 (4.4%) | |
| > 90 | 2,998 (36%) | 1,824 (44.7%) | |
| 61–90 | 4,960 (59.5%) | 2,074 (50.8%) | |
| **DBP mm Hg** | | | <0.0001 |
| < 60 | 430 (5.1%) | 324 (7.9%) | |
| 60–80 | 3,756 (44.4%) | 1,809 (44.2%) | |
| > 80 | 4,279 (50.5%) | 1,957 (47.8%) | |
| **SBP mm Hg** | | | <0.0001 |
| < 80 | 6 (0.1%) | 18 (0.4%) | |
| 80–120 | 1,924 (22.7%) | 979 (24%) | |
| > 120 | 6,542 (77.2%) | 3,090 (75.6%) | |
| **Pulse oximetry (percent)** | | | <0.0001 |
| 0–94 | 678 (8.2%) | 620 (15.2%) | |
| 95+ | 7,599 (91.8%) | 3,446 (84.8%) | |
| **Respiratory rate per minute** | | | <0.0001 |
| < 12 | 24 (0.3%) | 19 (0.5%) | |
| 12–20 | 7,802 (93.1%) | 3,481 (85.5%) | |
| > 20 | 551 (6.6%) | 573 (14.1%) | |
| **Injury/trauma, overdose/poisoning or adverse effect of medical/surgical treatment** | | | <0.0001 |
| Yes, injury/trauma | 2,585 (30.6%) | 567 (14.1%) | |
| Yes, overdose/poisoning | 83 (1%) | 56 (1.4%) | |
| Yes, adverse effect of medical/surgical treatment | 219 (2.6%) | 142 (3.5%) | |
| No | 5,385 (63.8%) | 3,156 (78.4%) | |
| Questionable injury status | 165 (2%) | 106 (2.6%) | |
| **Emergency severity index** | | | <0.0001 |
| Immediate | 85 (1.5%) | 117 (3.8%) | |
| Emergency | 721 (12.8%) | 760 (24.6%) | |
| Urgent | 2,820 (50.1%) | 2,002 (64.8%) | |
| Semi-urgent | 1,752 (31.1%) | 187 (6.1%) | |
| Non-urgent | 248 (4.4%) | 24 (0.8%) | |

| | IV fluid | | |
| | No | Yes | p value |
|---|---|---|---|
| **Medical history** | | | |
| Alzheimer's disease/Dementia | 113 (1.3%) | 93 (2.2%) | <0.0001 |
| Asthma | 917 (10.3%) | 485 (11.6%) | 0.028 |
| Cancer | 320 (3.6%) | 338 (8.1%) | <0.0001 |
| Cerebrovascular disease/History of stroke (CVA) | 330 (3.7%) | 252 (6%) | <0.0001 |
| Chronic kidney disease (CKD) | 313 (3.5%) | 324 (7.7%) | <0.0001 |
| Chronic obstructive pulmonary disease (COPD) | 511 (5.7%) | 404 (9.6%) | <0.0001 |
| Congestive heart failure (CHF) | 352 (3.9%) | 327 (7.8%) | <0.0001 |
| Coronary artery disease (CAD) | 509 (5.7%) | 522 (12.4%) | <0.0001 |
| Depression | 1,266 (14.2%) | 704 (16.8%) | 0.0001 |
| Diabetes mellitus (DM)—Type I | 52 (0.6%) | 50 (1.2%) | 0.0003 |
| Diabetes mellitus (DM)—Type II | 709 (7.9%) | 528 (12.6%) | <0.0001 |
| End-stage renal disease (ESRD) | 89 (1%) | 96 (2.3%) | <0.0001 |
| Pulmonary embolism (PE), DVT, or venous thromboembolism (VTE) | 176 (2%) | 105 (2.5%) | 0.058 |
| HIV infection/AIDS | 102 (1.1%) | 39 (0.9%) | 0.3111 |
| Hyperlipidemia | 993 (11.1%) | 759 (18.1%) | <0.0001 |
| Hypertension | 2,547 (28.5%) | 1,752 (41.8%) | <0.0001 |
| Obesity (BMI >= 30) | 673 (7.5%) | 430 (10.3%) | <0.0001 |
| Obstructive sleep apnea (OSA) | 251 (2.8%) | 203 (4.8%) | <0.0001 |
| Osteoporosis | 87 (1%) | 72 (1.7%) | 0.0004 |
| Substance abuse or dependence | 868 (9.7%) | 444 (10.6%) | 0.1336 |

**Note:**
The variables "Respiratory Rate," "Temperature," "Pulse Oximetry," "Heart Rate," "Payment Type," "Seen Within Last 72 Hours," and "Episode of Care" have missing data proportions ranging between 5% and 10%. The variables "Arrival Time," "Patient Residence," "Arrival by Ambulance," "Systolic Blood Pressure," "Diastolic Blood Pressure," and "Visit Related to Injury/Trauma, Overdose/Poisoning, or Adverse Effect of Medical/Surgical Treatment" have missing data proportions of less than 5%.

odds. Chronic conditions such as congestive heart failure, coronary artery disease, chronic kidney disease, diabetes, cancer, and hypertension significantly contributed to the likelihood of IV fluid administration. Conversely, factors associated with reduced likelihood included non-White race/ethnicity, residence in nursing homes or homelessness, and patients on Medicaid or uninsured. This detailed representation emphasizes clinically relevant predictors influencing IV fluid use, highlighting critical factors for clinical decision-making and resource optimization in emergency settings.

Figure 4 reports results for the LR models and Fig. 5 for GBC models, each using three NLP representations of the unstructured narratives—CountVectorizer, Word2Vec, and pre-trained GPT-2. Across both model families, combined models that integrated structured and unstructured data consistently outperformed models using either data type alone, regardless of the NLP technique. Among the NLP methods, clear performance differences emerged: in the GBC analyses (Fig. 3), the combined CountVectorizer model achieved the best AUC (0.786), followed by Word2Vec (0.785) and GPT-2 (0.772); the unstructured-only models showed the same ordering (CountVectorizer 0.735, Word2Vec

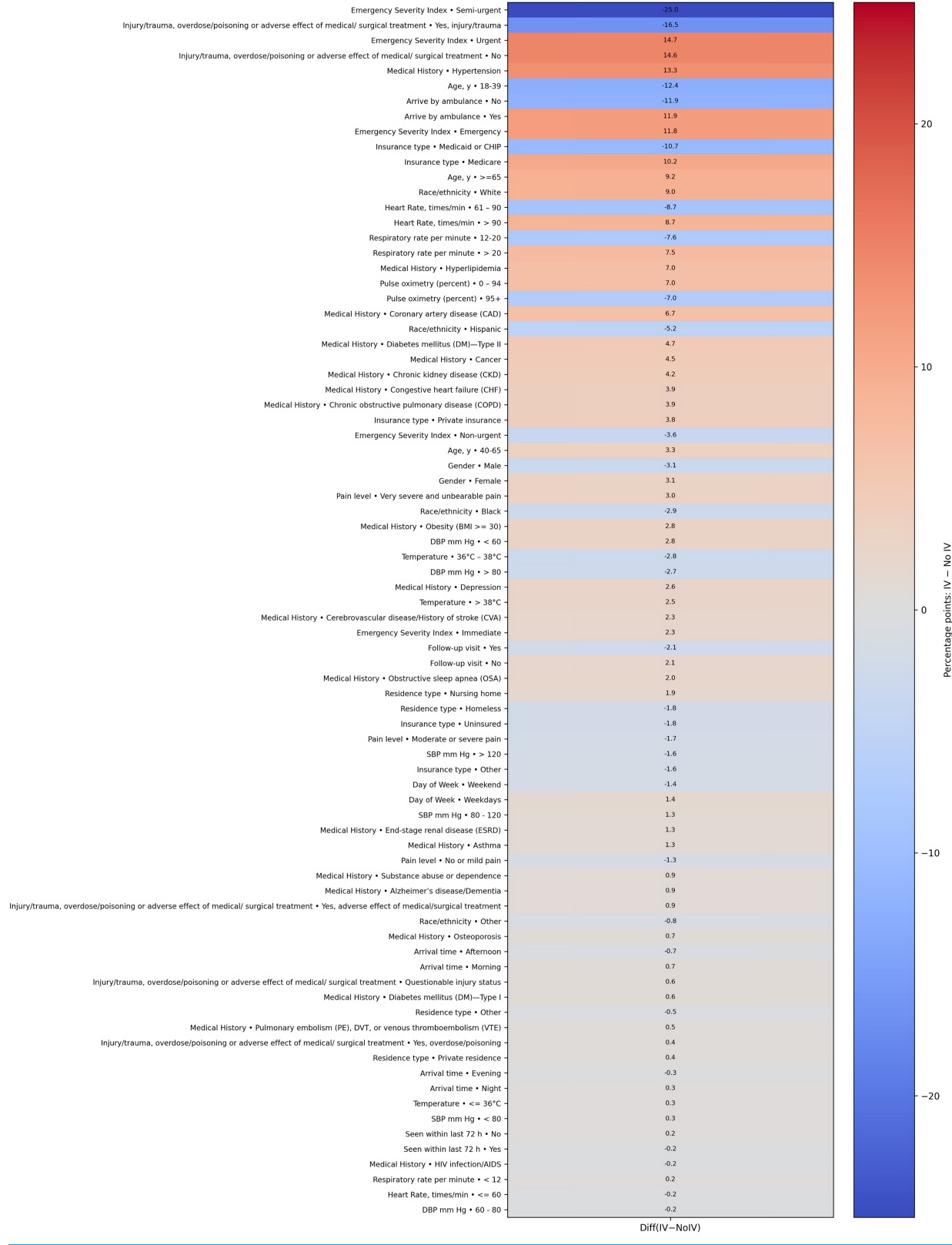

**Figure 2  Heatmap of group differences (percentage points).**

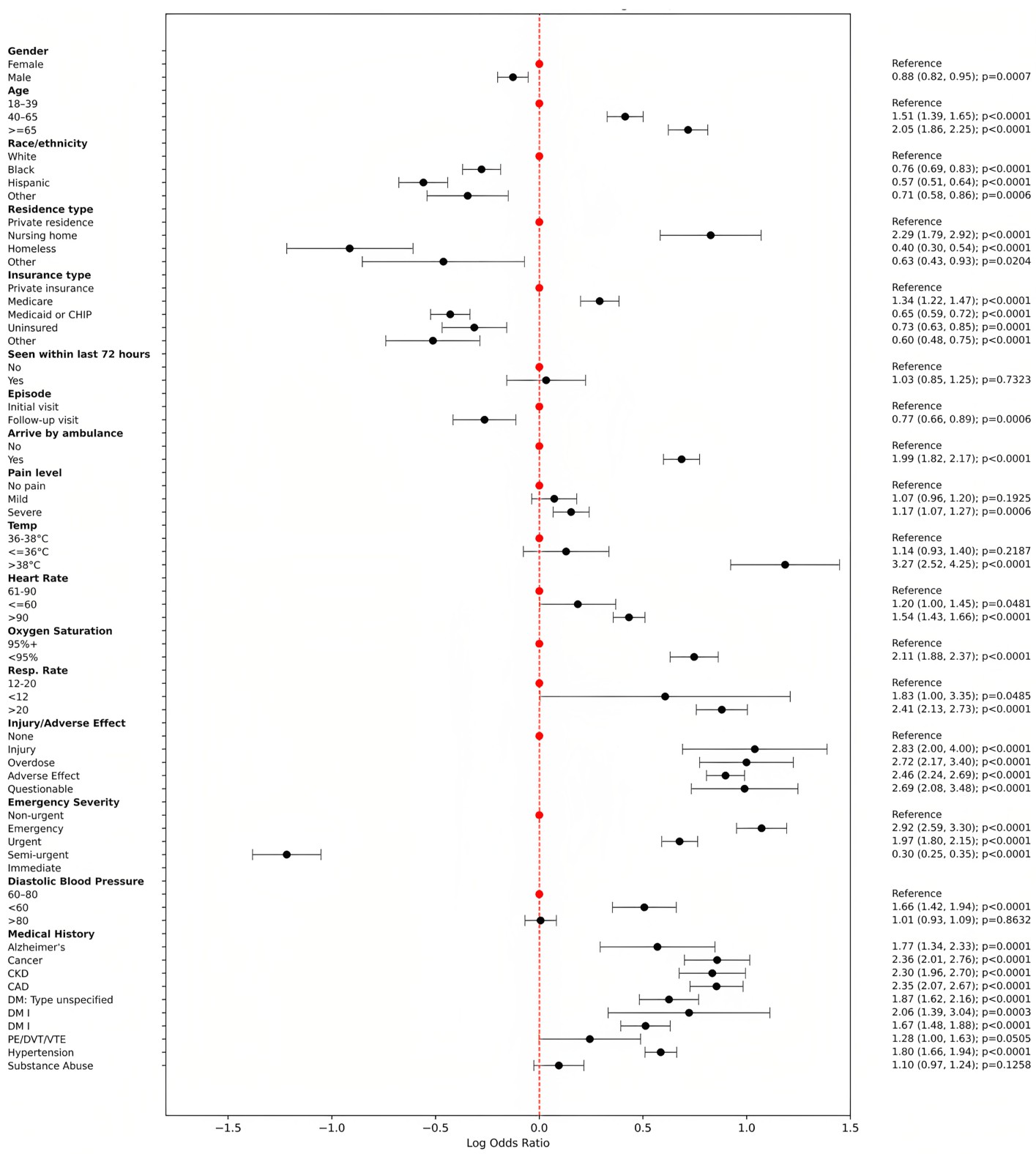

**Figure 3  Forest plot of odds ratios with 95% CI (log scale).**

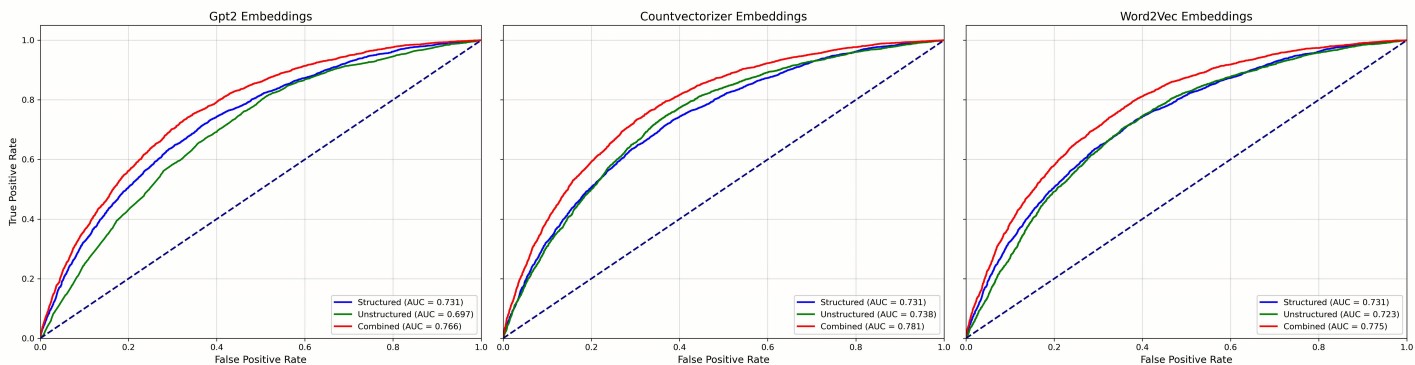

**Figure 4 Mean ROC curves for LR models predicting emergency department IV fluid utilization.**

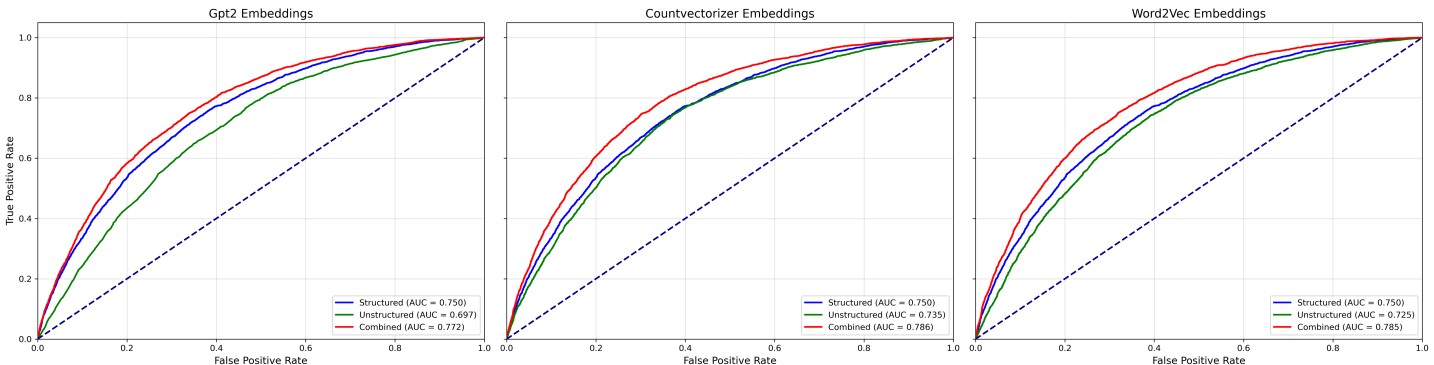

**Figure 5 Mean ROC curves for GBC models predicting emergency department IV fluid utilization.**

0.725, GPT-2 0.698). For LR (Fig. 2), the combined models followed the identical pattern (CountVectorizer 0.781, Word2Vec 0.775, GPT-2 0.766). Taken together, these results indicate that while incorporating patient narratives is critical, simpler keyword-based representations are more effective than a complex, context-based pre-trained model like GPT-2 for extracting predictive signal from the short emergency department chief-complaint texts.

## DISCUSSION

This study evaluated the predictive capability of integrating structured clinical variables with unstructured patient narratives to forecast IV fluid administration in ED settings. The primary finding confirms that an integrated, multimodal approach consistently outperforms models that rely on either structured or unstructured data alone. This aligns with emerging evidence supporting the fusion of diverse data sources to enhance the accuracy and efficiency of clinical decision-making. The improved performance underscores that patient narratives contain crucial, clinically relevant information not fully captured by traditional structured metrics (*Su et al., 2022*; *Zhang et al., 2019*, *2024*).

A key and unexpected finding of this study, however, emerged from our comparative analysis of different NLP techniques. Contrary to the prevailing trend of leveraging large, complex models, we found that traditional NLP methods, specifically the frequency-based CountVectorizer model, yielded superior predictive performance compared to the advanced, pre-trained GPT-2 transformer model (*Shaik et al., 2023*). This suggests that for the specific task of predicting IV fluid utilization based on short ED chief complaints, the choice of NLP methodology is not trivial and "more complex" does not necessarily mean "more effective".

Our primary finding that an early fusion of features provides a significant performance boost is consistent with previous work in the ED setting. For example, *Hong, Haimovich & Taylor (2018)* found that incorporating chief complaint text features with structured triage data significantly improved the prediction of hospital admissions. Our results, however, provide a crucial, nuanced counterpoint to the narrative that newer, more complex NLP models are always superior. While unexpected, this finding is not without precedent. In a comparison of transformer models and bag-of-words for classifying radiology reports, *Lu, Ehwerhemuepha & Rakovski (2022)* noted that the performance gain from a sophisticated Clinical-BERT model over simpler baselines was modest, highlighting that model complexity must be justified by the task. Our study extends this concept, demonstrating that for the concise, keyword-driven nature of ED chief complaints, the efficient keyword detection of a CountVectorizer model is more effective than the deep contextual understanding of a general-purpose transformer.

We hypothesize several reasons for this outcome. First, the nature of clinical text in the ED is typically short, factual, and keyword-driven (*e.g.*, "chest pain," "shortness of breath"). "Bag-of-words" models like CountVectorizer excel at capturing the presence and frequency of these critical keywords, which are themselves powerful predictive signals. Second, the strength of models like GPT-2 lies in understanding nuanced syntax and long-range context, which are largely absent in these brief patient complaints. Attempting to model deep context in short, factual text may introduce noise rather than improving signal detection. Finally, the GPT-2 embeddings used were from a general-purpose model, not fine-tuned on medical text. In contrast, CountVectorizer builds its vocabulary directly from the task-specific data, making its features highly relevant and optimized for this particular clinical dataset. Our findings thus highlight the critical importance of aligning model choice with data characteristics.

Despite the variations in NLP model performance, the key clinical predictors identified by our best-performing models remained consistent and clinically coherent. Factors such as arrival by ambulance, older age, a history of coronary artery disease, and elevated heart rate prominently emerged as significant predictors of IV fluid administration. The identification of these factors, which logically align with higher patient acuity, reinforces the clinical validity and applicability of the integrated modeling approach. This demonstrates that such models can reliably capture the key indicators that influence real-world clinical decision-making in emergency care.

The implications of this study for healthcare informatics are twofold. It reinforces the value of leveraging unstructured narrative data, but it also serves as a crucial reminder that the optimal NLP approach is task-dependent. For clinical prediction tasks involving short, keyword-rich text, simpler, more interpretable, and computationally efficient models may be superior to large, general-purpose language models. This methodological insight is valuable, suggesting that future research should focus not just on applying the newest models, but on rigorously comparing different techniques to find the best fit for specific clinical problems. This approach could be expanded to enhance predictive models for various other clinical outcomes and resource allocation challenges in healthcare.

## CONCLUSIONS

In conclusion, this study demonstrates that integrating structured clinical data with unstructured patient narratives substantially improves the accuracy of predictive models for IV fluid utilization in emergency departments. Our research further reveals that for the short clinical texts characteristic of emergency chief complaints, traditional NLP techniques like CountVectorizer are more effective at extracting predictive features than advanced, pre-trained GPT-2 models. This multimodal approach, combining the interpretability of logistic regression and the high performance of Gradient Boosting, offers a robust framework for enhancing clinical decision-making, streamlining resource allocation, and elevating patient care.

### Model justification

The selection of predictive models was guided by the dual requirements of clinical interpretability and high predictive accuracy, which are essential for developing trustworthy decision support tools in an emergency department setting.

**Logistic Regression (LR)** was chosen primarily for its transparency. In a clinical context, understanding *why* a model makes a certain prediction is often as important as the prediction itself. LR's straightforward parameter estimates allow clinicians to easily interpret the influence of each predictor on the likelihood of IV fluid administration. This level of interpretability is critical for building trust and facilitating the adoption of predictive tools in clinical practice, as it provides clear, actionable insights.

**Gradient Boosting Classifier (GBC)** was selected to maximize predictive performance. As a powerful ensemble method, GBC is highly effective at capturing the complex, non-linear relationships and interactions among variables that are common in heterogeneous clinical datasets. By leveraging its robust predictive capabilities, GBC serves as a benchmark for the high-end accuracy achievable on this task, ensuring that our performance evaluation is rigorous.

By employing this dual-model strategy, our study provides a comprehensive analytical framework. It balances the need for clinically actionable insights (from LR) with the goal of achieving high-fidelity, data-driven predictions (from GBC), thereby offering a more complete and practical solution for resource planning and clinical decision support in the ED.

## Limitation and future work

Several limitations should be acknowledged. First, the study's retrospective design and reliance on the NHAMCS-ED dataset may introduce biases inherent to observational data, including missing data and coding inaccuracies. However, rigorous data preprocessing and imputation methods were applied to mitigate these issues. Second, the model's generalizability should be validated through prospective studies and external datasets from diverse healthcare settings to ensure broader applicability and robustness. Future research should explore further refinements of NLP methodologies and investigate real-time integration of predictive models within electronic health records to support clinical decision-making actively (*Obermeyer & Emanuel, 2016*).

Evaluating the practical impact of these models on clinical outcomes and resource optimization through prospective, randomized controlled trials could provide additional evidence of their value in clinical practice. Third, our study employed an early fusion technique, where normalized structured clinical features and NLP-derived text features (from CountVectorizer, Word2Vec, or GPT-2) were concatenated prior to model training. While this method effectively integrated information from both modalities and demonstrated improved performance, it represents a relatively straightforward approach to multimodal data integration. Future work could explore more sophisticated fusion strategies, such as attention-based models or joint embedding methods, which might better capture the complex, synergistic interplay between structured and unstructured data. These advanced approaches could potentially offer enhanced robustness in handling common challenges inherent in EHR data, such as sparsity and irregularity.

### Funding

The authors received no funding for this work.

### Competing Interests

The authors declare that they have no competing interests.

### Author Contributions

- Hairong Wang conceived and designed the experiments, performed the experiments, analyzed the data, performed the computation work, prepared figures and/or tables, authored or reviewed drafts of the article, and approved the final draft.
- Haipeng Ling analyzed the data, performed the computation work, prepared figures and/or tables, and approved the final draft.
- Xingyu Zhang conceived and designed the experiments, performed the experiments, analyzed the data, authored or reviewed drafts of the article, and approved the final draft.

### Human Ethics

This study received exempt approval from the Institutional Review Board (IRB) at the University of Pittsburgh (protocol STUDY24120115).

## Data Availability

The code and raw data are available at GitHub:
https://github.com/VoVhw/GPT-2-IVF-prediction.

## Supplemental Information

Supplemental information for this article can be found online at http://dx.doi.org/10.7717/peerj-cs.3441#supplemental-information.

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
