# Peer review of "Integrating multimodal clinical data to predict intravenous (IV) fluid utilization: a comparative analysis of natural language processing techniques"

_PeerJ Computer Science, doi:10.7717/peerj-cs.3441_

## Round 0.1 · original submission · Major Revisions

· Academic Editor

Major Revisions

·

Basic reporting

The paper is overall clearly written with logical flow, the intro/background has provided sufficient context on the importance of predicting IV fluid utilization in EDs, and the potential for integrating unstructured patient narratives with structured clinical data, with the introduction section adequately stating the motivation for the study. And the manuscript is well-structured. However, the methodology is not clear -- there's no overall pipeline/framework illustration figure, and the method design, though sound, but some limited novelty (multi-modality encoding then late fusion).

Experimental design

As suggested in basic reporting, there's no overall pipeline/framework illustration figure, which is highly recommended to be included in the manuscript. The provided code implementation is fine; the evaluation methods, assessment metrics (AUC, accuracy, sensitivity, specificity, precision), and model selection methods (Logistic Regression, Gradient Boosting Classifier, late fusion for integrated models, 5-fold cross-validation) are adequately described. However, there seems to be no adequate baseline methods compared, limiting the paper's validity/superiority against other more advanced multimodal learning on structured & unstructured EHR data.
E.g,. An ICML paper: https://dl.acm.org/doi/10.5555/3618408.3620139 Improving medical predictions by irregular multimodal electronic health records modeling

Validity of the findings

The conclusions drawn are well-stated and appear to be directly supported by the presented results. The primary conclusion that integrated models (structured data + GPT-2 embeddings from narratives) outperform models using only structured or unstructured data is evident from the reported AUC values and other metrics. But as I suggested before, the paper could be much better with some more advanced methodology design in tackling multi-modal learning challenges (e.g., sparsity, irregularity, unpaired, etc.) with more methods to compare.

Reviewer 2 ·

Basic reporting

The article focuses on the advantages of using GPT-2 to improve the results of the supervised analysis conducted.

However, it does not provide an adequate comparison with similar studies in the literature.

In particular, a comparison should be made with similar studies that address the issue of textual data, providing the reader with an overview of the possible solutions. Indeed, there are several methods based on generative artificial intelligence that effectively tackle this type of problem. The authors should therefore provide the necessary tools to help readers fully understand the advantages of using generative AI.

Experimental design

The methods are described, but more detail should be provided regarding the rationale behind the choice of machine learning techniques used, as well as the specific advantages offered by the use of generative artificial intelligence.

Validity of the findings

To better justify their methodological choices and assess their level of innovation, the authors should also discuss similar studies in greater detail and present a comparative analysis of the results obtained using GPT-2 versus more established approaches, such as CountVectorizer, Word2Vec, etc. This would allow for a clearer understanding of the actual benefits brought by the use of GPT-2

The authors should at least compare their results with those obtained without using GPT-2.

Additional comments

Table 1 could be converted into an equivalent graphical representation to enhance clarity and visualize the data structure.

---

## Round 0.2 · accepted · Accept

· Academic Editor

Accept

Reviewer 2 has not re-reviewed this work. I have carefully checked the revisions and authors' responses against the reviewers' reports and can confirm that the authors' have addressed all the reviewers' suggestions.

Reviewer 1 mentioned lack of novelty and suggests to do more advanced methodology eg. multi-modal approach. However, novelty is not a requirement of the journal, but the work must show a contribution to the existing field.
The authors revisions have acknowledged this as a limitation. I commend the authors for ensuring reproducibility and implementing this work in Jupyter notebooks, and provenance in Zenodo. This manuscript is now ready for publication.

·

Basic reporting

no comment

Experimental design

no comment

Validity of the findings

no comment

Additional comments

no comment